# The Ribosomal Gene Loci—The Power behind the Throne

**DOI:** 10.3390/genes12050763

**Published:** 2021-05-18

**Authors:** Konstantin I. Panov, Katherine Hannan, Ross D. Hannan, Nadine Hein

**Affiliations:** 1PGJCCR and School of Biological Sciences, Queen’s University, Belfast BT9 5DL, UK; k.panov@qub.ac.uk; 2ACRF Department of Cancer Biology and Therapeutics, John Curtin School of Medical Research, Australian National University, Acton 2601, Australia; kate.hannan@anu.edu.au (K.H.); ross.hannan@anu.edu.au (R.D.H.); 3Department of Biochemistry and Molecular Biology, University of Melbourne, Parkville 3010, Australia; 4Oncogenic Signalling and Growth Control Program, Peter MacCallum Cancer Centre, Melbourne 3000, Australia; 5Department of Biochemistry and Molecular Biology, Monash University, Clayton 3800, Australia; 6School of Biomedical Sciences, University of Queensland, St Lucia 4067, Australia

**Keywords:** nucleolus, nucleolar associated domain (NAD), ribosomal genes, RNA polymerase I, transcription, heterochromatin, genome architecture, cell fate, differentiation, cancer

## Abstract

Nucleoli form around actively transcribed ribosomal RNA (rRNA) genes (rDNA), and the morphology and location of nucleolus-associated genomic domains (NADs) are linked to the RNA Polymerase I (Pol I) transcription status. The number of rDNA repeats (and the proportion of actively transcribed rRNA genes) is variable between cell types, individuals and disease state. Substantial changes in nucleolar morphology and size accompanied by concomitant changes in the Pol I transcription rate have long been documented during normal cell cycle progression, development and malignant transformation. This demonstrates how dynamic the nucleolar structure can be. Here, we will discuss how the structure of the rDNA loci, the nucleolus and the rate of Pol I transcription are important for dynamic regulation of global gene expression and genome stability, e.g., through the modulation of long-range genomic interactions with the suppressive NAD environment. These observations support an emerging paradigm whereby the rDNA repeats and the nucleolus play a key regulatory role in cellular homeostasis during normal development as well as disease, independent of their role in determining ribosome capacity and cellular growth rates.

## 1. Nucleoli and the rDNA Genes

Although genetic information is encoded in a linear DNA sequence, the transcription of particular genes (or gene clusters) depends on the surrounding chromatin structure and higher-order chromosomal interactions. Eukaryotic chromatin is tightly packed into the nucleus with a portion compressed into subnuclear domains, one of which is the nucleolus. Nucleoli form around ribosomal RNA (rRNA) genes (rDNA) and dictate the nucleolar morphology and the positioning of nucleolar-associated chromatin domains (NADs) within the nucleus. rRNA genes were first visualized in yeast in the late 1960s by Miller and Beatty using Miller spreads, which provided structural details of actively transcribed rRNA genes, specifically showing a single rDNA repeat transcribed by a multitude of RNA Polymerase I (Pol I) complexes, which they described as the Christmas tree structure [1]. These preparations further revealed that around 100 Pol I molecules simultaneously transcribe one gene at a speed of approximately 95 nucleotides/second [2]. In higher eukaryotes, the presence of histones in the transcribed region is a matter of debate, but it is widely accepted that the transcribed region is deprived of fully assembled nucleosomes [2,3], which are replaced by upstream binding factor (UBTF). The transcribed 47S precursor rRNA (pre-rRNA) is rapidly processed into the mature 28S, 5.8S and 18S rRNA, which assemble together with the 5S rRNA synthesized by RNA Polymerase III and approximately 79 ribosomal proteins translated from mRNAs transcribed by RNA Polymerase II (Pol II) into the 40S and 60S ribosomal subunits. While the process of ribosome biogenesis (RiBi) has long been associated with the nucleolus, more recently other essential non-ribosomal cellular functions have been attributed to this nuclear subdomain. The nucleolus is now recognized as a plurifunctional hub coordinating the nucleolar surveillance pathway in response to cellular stress [4,5,6,7,8], a modulator of genome architecture [9,10,11] and a phase-separated compartment for protein quality control [12].

## 2. Canonical and Non-Canonical rDNA Repeats

In humans, the rDNA genes are arranged in a head-to-tail orientation forming repeat arrays organized in the nucleolar organizer regions (NOR) at the short arm of the 5 acrocentric chromosomes. The precise organization and exact number of repeats is species, cell type and age dependent [13,14]. Canonical repeats in human cells are 43–45 kb in length and composed of a core and spacer promoter, a transcribed region containing both a 5′ and 3′ external transcribed spacer region (ETS), two internal transcribed spacer regions (ITS) and the 18S, 5.8S and 28S rRNA coding regions, with each individual repeat separated by a non-coding intergenic spacer (IGS; ~30 kb) [15] (Figure 1). In higher eukaryotes, the core promoter dictates transcription of the pre-47S rRNA [16,17], whereas the spacer promoter [18,19,20,21] mediates transcription of non-coding RNAs (see below). Transcription termination factor 1 (TTF-1) binds to the transcription terminator sites downstream of the 28S coding region and blocks Pol I elongation [22,23,24].

The upstream transcription enhancer elements (UTEEs), also known as the spacer promoter enhancer repeat, are another regulatory element located in the IGS [25]. This is the site of the formation of an enhancer boundary complex formed by CCCTC-binding factor (CTCF) and cohesion [24].

Upstream the enhancer boundary complex is flanked by nucleosomes, while downstream various components of the Pol I basal transcription apparatus were found, including Pol I, selectively factor -1 (SL-1), UBTF, RRN3, and TTF-1. The functional significance of these components is unclear as they are not involved in transcription. This is probably an artefact caused by the spatial proximity of the enhancer boundary complex and the core promoter, which leads to cross-linking of factors associated with the core promoter and the UTEE. It has been proposed that the enhancer boundary complex forms and serves as entry points for chromatin remodelling [24].

Traditionally, with exception of the IGS, which mainly contains repetitive sequences and transposable elements, the rRNA genes are believed to be highly conserved among species; however, increasing evidence suggest that not all rRNA genes are identical and instead exist in several variants [26,27,28]. The first report of a non-canonical rDNA repeat was by Caburet et al. in 2005, describing palindromic sequences arranged as mosaics with canonical repeats in human NORs [29].

The concept of heterogeneous rDNA repeats and copy number variation was further strengthened by two recent studies (Parks et al. and Wang and Lemos), which analysed whole genome sequencing data from the 1000 genome project [14] and different cancer types [13]. Parks and colleagues revealed that rDNA copy numbers varied greatly (about tenfold) between individuals within a human population and discovered pervasive inter- and intra-individual rDNA sequence variability. Interestingly, rRNA sequence variations are often associated with functionally important sites. For example, rRNA variations affecting inter-subunit bridge elements that establish the binding interface, linking the small and large ribosomal subunits impact on translation [14]. The same study described tissue-specific expression of rRNA variants in mice [14].

Analysis of over 700 tumours and corresponding normal tissues revealed a coupling of 5S rDNA array expansion with a loss of 45S repeats, which was most prevalent in TP53 mutant cancers. However, these variations were considered within the limits of natural variability and did not mediate an overall decrease in rDNA transcriptional output per cell. Two studies (Wang et al. and Ida et al.) hypothesized that loss of copies of 45S rDNA may be caused by replicative stress, as a result of rapid replication and high rDNA transcription rates affecting sister chromatid cohesion. They suggested that this loss would be beneficial for cancer cells as excessive 45S rDNA copy numbers are believed to promote genomic stability by facilitating recombinational repair [13,30]. The same study discussed the concept that loss of 45S copy numbers may be compensated for by epigenetic mechanisms controlling rDNA transcription. Thus, it is not surprising that loss of chromatin remodeller ATRX (α thalassemia/mental retardation X-linked), a member of the SWI/SNF family of helicase/ATPases, causes a substantial reduction in rDNA copy number [31].

In conclusion, genome-wide sequencing analysis [13,14] and genetic studies [31] suggest that the once considered conserved rDNA arrays are in fact one of the most variable regions of the genome.

## 3. Pol I Transcription Machinery

Three key “basal” Pol I transcription factors have thus far been identified; these are SL-1, Pol I associated regulatory factor RRN3 (also called TIFIA) and UBTF (also called UBF).

SL-1 is a complex of the TATA-binding protein (TBP) and 4 TATA-binding protein-associated factors (TAF), TAFI48 (TAF1A), TAFI63 (TAF1B), TAFI110 (TAF1C) and TAFI41 (TAF1D) [21,32]. SL-1 is responsible for promoter recognition [33] and together with UBTF, a high-mobility group (HMG) box protein (UBTF1 and UBTF2), drives the initial steps of pre-initiation complex (PIC) formation [34]. SL-1 binding to the CORE of the rDNA promoter is followed by binding of UBTF homodimers to both the CORE and upstream core element (UCE), which initiates rDNA promoter looping, bringing both regions into close proximity and thus stabilizing SL-1 [35,36,37]. How UBTF is recruited remains unclear; however, one study by van de Noebelen et al. suggested that CTCF may facilitate this process [38].

RRN3 has a unique HEAT repeat fold and regulatory serine phosphorylation sites [39]. RRN3 interacts with SL-1 to recruit the Pol I complex, the “core” Pol I subunits and auxiliary factors [40] to the promoter, thereby completing the formation of the PIC. After transcription initiation, RRN3 is released coinciding with the dissociation of the Pol I complex from promoter-bound initiation factors (promoter escape) [36,41,42,43].

UBTF not only acts as an archetypal transcription activator but also facilitates post-initiation events [36] via a variety of reported mechanisms. UBTF interacts with a heterodimer of two Pol I subunits, polymerase associated factor (PAF) 53 and CD3ε-associated signal transducer (CAST, also known as PAF49) to facilitate promoter escape [44]. In addition, UBTF regulates the elongation rate of Pol I by binding within the transcribed region of the rDNA where it forms so-called enhanceosomes, although this is still a matter of debate and awaits definitive structural analysis of UBTF on the rDNA [35]. In addition, UBTF binding affinity to the rDNA is regulated by differential phosphorylation [45,46,47,48].

More globally, UBTF1 has been reported to remodel chromatin, such as converting inactive rRNA genes into active chromatin by displacing linker histone 1 histones [24,49,50], as their binding is mutually exclusive, while UBTF2 exhibits no chromatin remodelling activity. Thus, UBTF1 is critical for the topological organization of the rDNA repeats. UBTF is also critical during mitosis as it remains constitutively bound to the rDNA pausing transcription; thus, it is thought to ‘bookmark’ active genes to facilitate re-initiation of transcription when cells enter the interphase [51]. This mitotic bookmarking enables UBTF to play a central role in the formation of NORs and maintaining secondary constrictions at active NORs [52,53]. Based on the UBTFs’ extensive binding across active genes and their absence in the IGS (except for the enhancer and promoter regions), it is considered a mark of euchromatic rDNA.

Around two decades ago, two distinct Pol I subcomplexes were identified initially in yeast and later in humans (Pol Iα and Pol Iβ) [40,41]. Both subcomplexes were catalytically active, but only one representing a small proportion of total cellular Pol I (2–10%) was competent to initiate transcription from the rDNA promoter. The competence to initiate transcription was determined by the presence of RRN3. The subcomplexes consist of 12–13 so-called “core” subunits but associate with a different set of auxiliary factors that further determine the functionality of these complexes. For example Pol Iα, which is the elongating form of Pol I, contains histone chaperone FACT (facilitates ATP-independent nucleosome remodelling) [54], whereas Pol Iβ, which is the initiating form of Pol I, is defined by the presence of RRN3, DNA topoisomerase (Top) IIα and casein kinase II [40,55,56,57,58,59]. As Pol I starts transcribing, the resultant nascent rRNA immediately associates with the pre-RNA processing machinery, tightly coupling rRNA synthesis and maturation [60]. In addition to the above “core” components of the Pol I transcription machinery, a multitude of other factors, such as nucleolin, nucleophosmin, Top I/II and chromatin-remodelling complexes [58,61,62], have been implicated in the regulation of Pol I loading and elongation. Finally, as mentioned above, termination of Pol I elongation and its dissociation is mediated by TTF-1 binding to the terminator elements [63].

## 4. Regulation of Ribosomal Gene Transcription

Pol I transcription is regulated in response to extracellular (e.g., environmental stimuli, stresses) or intracellular (e.g., cell cycle, cell growth) stimuli by a number of mutually non-exclusive mechanisms, including post-translational modification of the Pol I machinery, alterations in rDNA topology, changes in rDNA chromatin structure and through non-coding RNAs [64].

In response to environmental stimuli (e.g., growth factors) and various stresses (e.g., DNA damage), components of the Pol I transcription machinery are targeted by a number of signalling pathways, including PI3K-mTOR, RAF-MEK-ERK, AMPK and AKT signalling, which often converge, forming a complex signalling network [65,66,67,68,69,70,71,72,73]. Importantly, the severity and duration of cellular stresses (e.g., starvation, DNA damage, heat shock) induce differential activation of these pathways and alter the epigenetic landscape of the rDNA via chromatin modifiers and remodellers, leading to the changes in chromatin accessibility [74,75,76,77]. Apart from Pol I-specific transcription factors (TF), a range of other TF’s and oncogenes, previously only associated with Pol II-dependent transcription, have now been reported to also modulate rRNA synthesis. For example, acute myeloid leukemia (AML) 1-ETO, an AML-specific fusion-protein, has been reported to bind to human rRNA genes and promote Pol I transcription in malignant myeloid cells [78], while AML1 (Runx1) downregulates Pol I transcription [78].

In general, DNA topology is affected by transcription and replication. Top are a family of enzymes that release torsional stress at transcribed and replicated DNA loci. Two types of Top, Type I and Type II, relax supercoiled DNA by catalysing either single-strand or double-strand DNA breaks, facilitating DNA rotation or the passage of one DNA strand and re-ligation.

Experiments in yeast showed that Top I is involved in alleviation of the negative superhelical density formed behind elongating Pol I, whereas Top II is required for resolving positive supercoiling formed ahead of the transcription complex [79].

More recently, Denissov et al. revealed that regulatory elements, including the promoter, upstream region and terminator of actively transcribed genes, spatially interact throughout the cell cycle, forming so-called core–helix structures and Top I plays an essential role in maintaining this topology [80]. This rDNA loop formation brings the initiation and termination sites into close proximity, which was suggested as a means to facilitate the ‘recycling’ of Pol I complexes. A number of factors, including TTF-1 and c-Myc, have also been identified as being involved in this loop formation. Direct evidence of active rDNA looping was recently presented by Maiser and colleagues [81]. Ray et al. showed that, in human cells, Top IIα altered the rDNA topology at the rRNA core promoter and this was required for the assembly of functional PICs [58].

Non-coding RNAs were first described as regulators of rDNA transcription over a decade ago [82], specifically as being critical for rDNA silencing by facilitating the interaction between the nucleolar remodelling complex (NoRC) subunit TTF-1 interacting protein (Tip5) and TTF-1 [27,83]. These non-coding RNAs are proposed to promote heterochromatin formation at the rDNA and other chromosomal repeats [84,85] (for recent reviews, see [86,87]).

Interestingly, Abraham et al. proposed recently that Pol II-driven production of anti-sense transcripts originated from the rDNA IGS facilitated the formation of DNA–RNA hybrid structures, known as R-loops, at the boundaries of the IGS and coding rDNA regions. R-loops prevented Pol I-driven transcription of the IGS and also the production of sense intergenic noncoding RNAs (sincRNAs) that can negatively affect rRNA transcription [88]. These findings provide a potential direct mechanism that couples both Pol II and Pol I transcription activities.

## 5. rDNA Chromatin Dynamics

It is well established that chromatin undergoes extensive and dynamic remodelling in order to modulate gene transcription. While repositioning and remodelling of nucleosomes, plus modifications of histones at specific sites (e.g., promoters), are central to the control of gene transcription, earlier studies in yeast and *Drosophila* oocytes suggested that actively transcribed rRNA genes were nucleosome depleted [89,90]. However, this view was challenged by another chromatin study in yeast revealing that active rRNA genes were indeed associated with histones and nucleosomes [91]. Currently, there is no consensus as to whether in higher eukaryotes actively transcribed rDNA are associated with functional nucleosomes or core histones, such as those typically located on Pol II-transcribed genes. A number of studies have demonstrated that histones H3 and H4 are associated with transcribed rDNA in human cells [74,92], while other studies disagree [24,87].

Integrative genomic analysis of human and mouse embryonic stem cells (ESCs) revealed similarities in the enrichment of euchromatic and heterochromatic histone marks ~2 kb upstream of the rDNA core promoter, while the other regions within the rDNA were markedly different [93,94]. A recent study by Herdman and colleagues reported active histone marks (H3K4me2/3, H2A.Z/ac, H3K9ac, H3K27ac and H3K36me3) exclusively in the enhancer region but not within the transcribed region or the IGS upstream of the enhancer boundary complex [24]. Interestingly, in nutrient-starved cells, histone H3 was found associated with the transcribed regions of active rDNA repeats, but this was rapidly removed when the cells recovered from starvation [74], highlighting the dynamic nature of rDNA chromatin.

A landmark study in 2008 presented evidence that, in fact, the rDNA repeats exist in three distinct chromatin configurations: (i) transcriptionally active rDNA repeats that are hypo-CpG methylated at the promoter-specific CpG, enriched for euchromatin histone marks and bound by UBTF; (ii) pseudo-silent/poised rDNA repeats that are hypo-CpG methylated at the promoter, bear repressive histone modifications, but are not bound by UBTF, and thus exist in a closed chromatin conformation; and (iii) silent rDNA repeats that are promoter hyper-CpG methylated and associated with heterochromatic histone marks adopting a highly compact chromatin state [50]. The silencing of the rDNA repeats is promoted by NoRC, a complex which recruits histone and DNA modifiers such as histone deacetylases 1 (HDAC1) and DNA methyltransferase 1/3a (DNMT1/3a) to the rDNA [76].

In contrast to the promoter-associated CpG methylation, the presence of CpG methylation within the rDNA coding region and its association with active transcription is still a matter of debate and complicated by their repetitive multi-copy nature. A recent study by Wang and Lemos reported that the rDNA CpG hypermethylation strongly correlated with aging, thus serving as an evolutionarily conserved biological clock [95]. However, no link between age-mediated methylation and the level of rRNA transcription has been reported.

The transcriptional pseudo-silent/poised rDNA chromatin conformation is established by nucleosome remodelling by deacetylation complexes (e.g., NuRD [96] and eNoSc [75]). While these poised rDNA promoters are unmethylated and have nucleosomes positioned to prevent transcriptional initiation, they are often characterized by either bivalent histone modifications (e.g., H3K4me3 and H3K27me3) or fully repressive histone marks (e.g., H3K4me1/2 and H3K9me3). Re-activation of transcription in this case will depend on the re-positioning of a nucleosome at the rDNA promoter by the DNA-dependent ATPase Cockayne syndrome protein B (CSB) [96] and histone modifications by the coordinated actions of histone methyltransferases (e.g., MLL1–2 [97]), histone demethylases (e.g., PHF8 [98], KDM4A [74] and G9a [99]) and histone acetyltransferases (e.g., PCAF [100]). It is feasible to propose that these enzymes are not acting alone but are in fact part of a large activating complex. This idea is supported by data showing that many chromatin modifiers involved in the activation of rDNA transcription interact with the scaffold protein WD repeat-containing protein (WDR)5 [101,102].

Despite acknowledged species-specific differences in nucleosome occupancy at actively transcribed ribosomal genes, this remains a “hot-topic” of discussion. To add to the epigenetic complexity of rDNA chromatin, a number of non-canonical histone variants have been identified that bind to the rDNA. For example, histone H3.3, a variant previously described to be involved in transcriptional activation as well as gene silencing, was recently demonstrated to bind to the rDNA [31]. Other histone variants, e.g., H2AZ, was located on the IGS region of the rDNA and incorporated into the rDNA under high glucose conditions [103,104]. Interestingly, phosphorylated H1.2 and H1.4 variants of histone H1, which are commonly associated with inactive rDNA chromatin, were found to promote transcription of the rDNA genes [103,105].

In addition to histones and Pol I-specific TFs, there are ubiquitous DNA-binding proteins, such as CTCF, and structural maintenance of chromosome (SMC) complexes, such as cohesin and condensin [38,106,107], which can bind and modulate the epigenetic state of the rDNA genes. CTCF binds upstream of the rDNA spacer promoter and interacts with UBTF and other Pol I complex components, suggesting it is a regulator of Pol I transcription and rDNA chromatin. This was further supported by the finding that CTCF depletion reduced UBTF and Pol I binding near the spacer promoter [38]. Although these proteins are not considered components of the core transcription machinery, they are integral to the structure of rDNA chromatin [24,87,108].

Overall, regulation of rDNA chromatin structure plays a pivotal role in maintaining the balance between actively transcribed, pseudo-silenced and silenced rDNA repeats, and thus is crucial for controlled responses to, e.g., stress, development, aging and genomic instability. Despite the identification of a number of key players, our understanding of these complex mechanisms, and in particular the crosstalk between them, is still limited, and the focus of ongoing research.

## 6. Role of the Nucleolus in Spatial Genome Organization and Pol II Transcription

The overall three-dimensional (3D) organization of the genome highly depends on the formation of chromatin contacts within and/or between each chromosome and the nuclear domain. These types of interactions include: (i) chromatin loop formation that bypass long genomic distances and connect distant genomic regions, such as enhancers and promoters; (ii) the formation of topologically associated domains (TADs) that are local-interacting DNA neighbourhoods; and (iii) regions where genomic loci interaction is driven by their transcriptional activity.

Several studies have demonstrated a direct role for the nucleolus in genome organization and as a global modulator of all transcription (Pol I, II and III) [9,10,11,109,110,111]. The nucleolar periphery contains satellite DNA repeats that form a perinucleolar heterochromatic dense shell. Specific genomic regions located in the perinucleolar region are called nucleolus-associated domains (NADs) (for a recent review, see [110]), whereas those associated with nuclear lamina are called LADs; however, in humans, a proportion of NADs and LADs overlap (Figure 2). NADs were first described in 2010 by two groups, van Koningsbruggen et al. and Nemeth et al., who demonstrated that specific genomic sequences interact with the nucleolus (around 4% of the genome) [9,11]. Using complementary approaches of fluorescence in situ hybridization (FISH) and DNA sequencing, they described the NAD-interacting genomic domains as being generally located in gene-poor regions but did identify a number of specific gene families, including zinc-finger, olfactory receptor and defensin genes, as well as satellite pericentromeric and centromeric repetitive sequences, regions of the inactive X-chromosome (Xi), as being enriched with NADs. Further, some tissue-specific expressed gene clusters were reported as associated with NADs, including two immunoglobulin clusters and T-cell receptor genes [9,11]. Analysis of RNA coding genes revealed an enrichment of Pol III-dependent 5S and transfer RNA genes in NADs, suggesting that their spatial organization within the nucleus may play a key role in their transcriptional regulation. This idea was further supported by the finding that NADs are enriched in repressive histone marks, such as H3K27me3, H3K9me3 and H4K20me3, while active marks are excluded, which correlated with a decrease in global gene expression of NAD-associated loci [9]. In fact, NADs are nuclear territories with their proposed primary function being the maintenance of heterochromatin at interacting regions. For example, the Xi continuously revisits the nucleolus during cell cycle progression through S phase to maintain its heterochromatic state [112].

However, NADs are not solely enriched on transcriptionally silenced genes; a number of highly transcribed Pol III-dependent RNA genes are also NAD associated. The dynamic regulation of the NAD interactions with genomic loci still requires intensive study. Comparative Hi-C analysis of NAD interactions in human embryonic fibroblasts revealed that surprisingly most of these interactions remained unchanged when comparing proliferating and senescent cells, with the exception of specific satellite sequence clusters that segregate from the nucleoli after cells underwent replicative senescence [113]. To shed more light onto the dynamic nature of NAD interactions and whether those interactions may contribute to disease development, Diesch and colleagues mapped genomic loci interacting with the nucleolus during malignant transformation [10]. Remarkably, the study demonstrated that Myc-driven malignant transformation of B-cells is associated with a significant increase and reorganization of rDNA-NAD contacts due to activation of previously silent rDNA genes (rDNA class switching). This spatial rearrangement correlated with gene expression changes at associated genomic loci, impacting the Pol II-transcribed gene ontologies, including B-cell differentiation, cell growth and metabolism, changes that contribute to malignant cell fitness [10]. Moreover, these studies support, for the first time, a model where structural changes in rDNA chromatin and subsequent rDNA-NAD reorganization promote gene expression changes at associated loci, which influence clonal selection of a malignant cell population (Figure 2).

Interestingly, most of the developmental Hox gene clusters rarely associated with the rDNA, supporting the idea of a highly coordinated framework of spatially and temporally defined genomic interactions [114]. A recent study by Vertii et al. revealed an even more complex situation, describing two functionally distinct classes of NADs in mouse embryonic fibroblasts. Heterochromatic Type I NADs were associated with low transcriptional activity and late DNA replication, and linked with both the nucleolar periphery and nuclear lamina. In contrast, Type II NADs exhibit higher Pol II-dependent gene transcription regulating genes important for development, earlier replication and were exclusively localized to the nucleoli and not the nuclear lamina [109]. The discovery of distinct types of NADs was further supported by a recent study that identified two NAD subtypes in mouse ESCs [115]. Although earlier studies suggested a stochastic recruitment of heterochromatic loci to either the nuclear lamina or nucleolar periphery, the dynamic nature and trans-acting factors, such as lamin B receptor and chromatin assembly factor 1 subunit p150, which modulate LAD and NAD localization, suggests that at least some loci are specifically recruited to one or the other [116,117]. The underlying mechanism of how NADs are recruited to the nucleolar periphery and how these interactions are dynamically regulated is still unknown; however, several studies imply that phase separation plays an important role [118].

Therefore, the number and size of nucleoli, which depends on the rate of Pol I transcription, can drive changes in the spatial organization of the genome, which directly modulates the global transcriptome. These findings indicate another layer of 3D transcriptional regulation, potentially playing an important role during normal cellular processes such as differentiation and aging, but may also, if dysregulated, be associated with malignant transformation.

## 7. Pol I Transcription in Differentiation and Development

The first link between Pol I transcription and differentiation dates back to the 1940s where high levels of rRNA synthesis and large nucleoli were observed in stem cells while rates attenuated as cells differentiated. These observations were further confirmed by subsequent studies, e.g., Altmann and Leblond revealed that nucleolar morphology and size changes were associated with a loss in rRNA synthesis when columnar cells matured and migrated from the crypt to the villus in rats [119]. Consistent with the decline in Pol I transcription, the level and/or activity of SL-1, UBTF and other RiBi-associated factors decrease upon differentiation, accompanied with changes in rDNA chromatin [120,121,122,123,124,125,126,127,128,129].

Another line of evidence supporting the notion that rDNA transcription is directly involved in the regulation/maintenance of pluripotency and differentiation is the fact that canonical pluripotency factors, such as Oct3–4/POU5F1, Nanog and Klf4, bind to the regulatory and transcribed regions of rDNA in mouse ESC, and this was also confirmed for Oct3–4/POU5F1 [94] in human ESC. In parallel to the loss of pluripotency TF binding at the rDNA, lineage commitment factors C/EBPbeta, MyoD and Mng TFs are recruited to the rDNA and suppress rDNA transcription during differentiation [130].

Thus, for many years, the decrease in the level of rRNA synthesis was regarded as a consequence of differentiation due to reduced metabolic requirements rather than being a driver. This view is now challenged by an increasing number of studies showing that inhibition of Pol I transcription can drive cell differentiation. A study by Hein et al. revealed that pharmacological inhibition of Pol I in murine models of AML and human AML cell lines led to an increase in apoptotic cell death, a delay in cell cycle progression and the induction of myeloid differentiation in leukemic blasts [131]. In line with these findings, Hayashi et al. showed that knockdown of TIF-1A (RRN3) induces differentiation in normal mouse haematopoietic stem cells (HSC). Interestingly, mouse HSC have moderate levels of rRNA transcription that progressively increases in lineage-committed progenitors while rRNA transcription levels are low in terminally differentiated cells [132].

Prakesh et al. described the dynamic regulation of rRNA transcription during epithelial–mesenchymal transition (EMT) in breast cancer. While rRNA transcription and processing as well as RiBi-associated factors, including UBTF, RRN3 and nucleolin, are transiently increased during EMT, mesenchymal cells have reduced Pol I transcription. The elevation of rRNA transcription observed during EMT resulted from previously silenced rDNA repeats being activated by the release of TIP5 from the rDNA promoter, a decrease in rDNA promoter methylation and increase in H3Kme3 and H3K27ac. Interestingly, Snail1, a key regulator of EMT, is also recruited to the rDNA while cells undergo EMT [133]. Pharmaceutical inhibition of Pol I transcription halts EMT and induces tumour cell differentiation to a non-invasive, luminal phenotype in MMTV PyMT mice [133]. Other compelling evidence were observed in *Drosophila* ovarian germline stem cells (GSCs). GSCs are enriched for RiBi factors and display high levels of Pol I transcription compared to their differentiating daughter cell. Depletion of the *Drosophila* SL-1-like complex subunits under-developed (Udd) and TAF1B, resulted in reduced GSC proliferation and GSC loss. In contrast, overexpression of TIF-IA (RRN3) increased rRNA synthesis and impaired GSC differentiation [127].

Understanding how Pol I transcription and differentiation are mechanically linked is critical, as both are inherently important for normal development but are frequently (but not universally) altered in cancer. The observation that Pol I transcription inhibition induces differentiation gives rise to the speculation that elevated rates of rRNA transcription may cause an opposite effect and induce (or at least contribute) the differentiation defect observed in malignancies. Further, it would explain why inhibition of Pol I restores a cancer cell’s ability to undergo differentiation. How the rate of rRNA synthesis affects cell fate requires further investigation. A possible mechanism may involve changes in long-range rDNA–NAD interactions caused by changes in the rate of Pol I transcription. As the level of transcription determines nucleolar size and morphology, and thus the sub-nucleolar organization of rRNA genes, changes in the rate of Pol I transcription will alter these and subsequently reorganize the rDNA–NAD interactome, potentially changing Pol II-dependent gene expression programs associated with pluripotency and differentiation.

Collectively, these findings provide strong evidence that rRNA synthesis is critical for stem cell maintenance, and its downregulation can drive differentiation.

## 8. How the Nucleolus and Ribosomal Genes Maintain Genome Stability

Over the last decade, our concept of the nucleolus has emerged from being simply the site of RiBi to a multifunctional hub. Studies of the nucleolar proteome provided the first evidence by revealing that, of the total number of ~4500 proteins sequestered in the nucleolus, a large percentage was not associated with RiBi. For example, a significant number of DNA-damage response (DDR) proteins, including ~130 DNA repair proteins [134,135,136], are enriched in the nucleolus. Whether the nucleolus is their site of action or it primarily serves as a deposition site remains under investigation; it is clear, however, that in response to DNA damage some nucleolar proteins and importantly the rDNA relocates to specific sub-nucleolar domains, e.g., nucleolar caps, promoting the idea that the nucleolus may coordinate the DDR. Interestingly a number of studies have also now revealed that some DNA repair proteins, including APEX1, BLM, HUWE1, RPS27A/eS31 and BRCA1, have additional functional roles in RiBi [134,137]. In addition, some RiBi factors are also involved in DNA repair, such as the nucleolar protein Treacher Collins syndrome protein 1 (TCOF1), which regulates Pol I transcription and is mutated in Treacher Collins syndrome [138]. A role for TCOF1 in DDR has been reported by two groups, Larsen et al. and Ciccia et al., demonstrating that the Nijmegen breakage syndrome protein 1 (NBS1)-TCOF1 complex mediates silencing of ribosomal gene transcription and also transiently binds to DNA double-stranded breaks (DSB) [139,140]. Interestingly, NBS1-dependent recruitment of DNA Top II-binding protein 1 (TOPBP1) is mediated via the ATM/ATR signalling pathway targeting the nucleolar phosphoprotein Treacle. This triggers inhibition of rRNA synthesis and nucleolar segregation in response to rDNA breaks [141]. Furthermore, the histone demethylase JMJD6, which is rapidly recruited to nucleolar DNA damage sites, interacts with Treacle and modulates its interaction with NBS1. JMJD6 is dispensable for rDNA transcription, but is crucial for the relocalization of rDNA to the nucleolar caps [142]. Thus, cooperation between the DNA repair machinery and RiBi factors is likely to underpin the cellular response to DNA damage, thus demonstrating a role for the nucleoli in maintaining genomic stability.

rDNA genes are considered genomic hotspots of recombination; their repetitive nature and high transcription rate make these repeats highly susceptible to replicative stress and replication fork stalling, causing DNA DSBs [143]. As mentioned above, ATRX-depletion in mouse ESC drives the selective loss of rDNA copies and thus is proposed to contribute to tumorigenesis through rDNA instability [31]. Interestingly, studies investigating the effect of rDNA copy variation in *Saccharomyces cerevisiae* revealed that loss of rDNA copies leads to a disequilibrium between active and inactive rRNA genes. Consistent with the findings in *Saccharomyces cerevisiae*, loss of mammalian Sir2-homolog SIRT7 induces excision and loss of rDNA copies, promoting rDNA instability and subsequent cellular senescence in primary human cells [144]. The NAD^+^-dependent protein deacetylase SIRT7, a chromatin-silencing factor, is required for the stabilization of components of the NoRC to the rDNA and therefore is crucial for heterochromatin formation [144]. Alternatively, silent rRNA genes are thought to be important for DNA recombination repair, NAD formation and, as a result, rDNA stability.

Not only do rDNA copy number variations impact genome stability, a number of rDNA-associated factors have been reported to play a role in the maintenance of rDNA/genome integrity. Recent findings revealed that UBTF depletion leads to DNA damage, abnormal mitotic progression and formation of micronuclei. However, these effects on genome stability are not simply mediated by a reduction in rDNA synthesis since knockdown of RRN3 equally reduced Pol I transcription but failed to induce micronuclei formation [145]. Another rDNA-specific factor indispensable for the orderly coordination of ribosomal gene transcription and rDNA replication is TTF-1. Recent evidence suggests that TTF-1 not only stops the elongating polymerase but also serves as a replication fork barrier protein, preventing the transcription and replication machineries from colliding [146].

Another structure thought to threaten genome stability is the R-loop. R-loops are RNA:DNA hybrids that form naturally when genes are actively transcribed, such as the ribosomal genes that have high rates of transcription and are GC-rich regions, both of which favour their formation [147]. R-loops can substantially impact genome integrity and can be a curse as well as a blessing, as these hybrid structures are proposed to regulate gene expression and facilitate homologous recombination (HR), which supports rDNA stability/integrity [148,149]. On the other hand, R-loops are suspected to pose an impediment to the DNA replication machinery during S-phase with deleterious consequences for genome fidelity. R-loops are discussed as being a major factor responsible for transcription–replication collision causing DNA breaks and subsequent genome instability. A number of human neurological disorders, such as Huntington and myotonic dystrophy type 1, have been linked to mutations in genes related to R-loop formation/metabolism, e.g., R-loop induced DNA damage at the rDNA [147]. Other higher-order structures linked to genomic instability and biological process, such as replication, transcription, translation and epigenetic regulation, are G-quadruplexes (G4), where four-stranded guanine-rich structures form in DNA and RNA, including the rDNA repeats. Aberrant G4 formation within the genome is implicated in human diseases such as cancer and receives in this context increasing interest as a druggable target [150].

Overall, it is conceivable that relocalization of rDNA to nucleolar caps as a result of DNA damage and/or Pol I inhibition, or alteration in rDNA copy numbers as a result of loss/mutation of some proteins, may affect the number and distribution of NADs. Such alterations of the NADs can lead to changes in the cellular transcriptome, which provides an explicit link between the role of rDNA loci in maintaining genomic integrity and controlling cell fate.

## 9. Conclusions

The discovery of the plurifunctional nature of the nucleolus about two decades ago was rightly regarded by the scientific community as a paradigm shift, and since then our knowledge of the functions of the nucleolus and our understanding of the underlying molecular mechanisms have expanded significantly. It has become clear that the nucleolus plays a key role in various cellular processes, such as carcinogenesis, development, differentiation and aging. Recent studies indicate that the rDNA loci and Pol I transcription are important for the dynamic regulation of global gene expression via modulating long-range genomic interactions with the suppressive NAD environment. Research also demonstrates a role for the rDNA chromatin structure in the spatial organization of the genome and genome stability. However, there are many gaps in our understanding of the functions and structure of the nucleolus. Questions about how heterogeneous the rDNA is, whether rDNA heterogeneity manifests in heterogeneous ribosomes, what is the mechanism for selecting the rDNA copy number, and how is the ratio between active and inactive repeats selected and maintained, are just a few still to be satisfactory answered. Moreover, despite our new understanding of a range of functions, to which the nucleolus is pivotal; a picture as to how these functions are interrelated is still unclear. It is evident that further intensification of research aimed at a comprehensive understanding of the role of rDNA and the nucleolus, including its impact on cell fate is required. In more practical terms, it is tempting to speculate that, given the plurifunctionality of the nucleolus and its dependence on ongoing rDNA transcription, the efficiency of the recently described Pol I inhibitors to treat cancer may be mediated in part through their effect to disrupt the non-ribosomal functions of the nucleolus.

## Figures and Tables

**Figure 1 genes-12-00763-f001:**
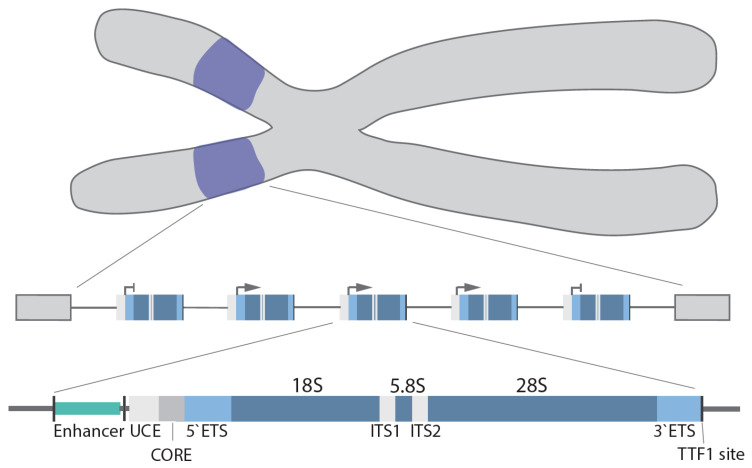
rDNA gene arrays (purple) are located on the short arms of the human acrocentric chromosomes. Organization of a single rDNA gene: enhancer, upstream control element (UCE), core promoter (CORE), 5′/3′ external transcribed spacer (ETS), 18S, 5.8S, 28S, internal transcribed spacer (ITS1/2), and transcription terminator factor 1 (TTF-1) site.

**Figure 2 genes-12-00763-f002:**
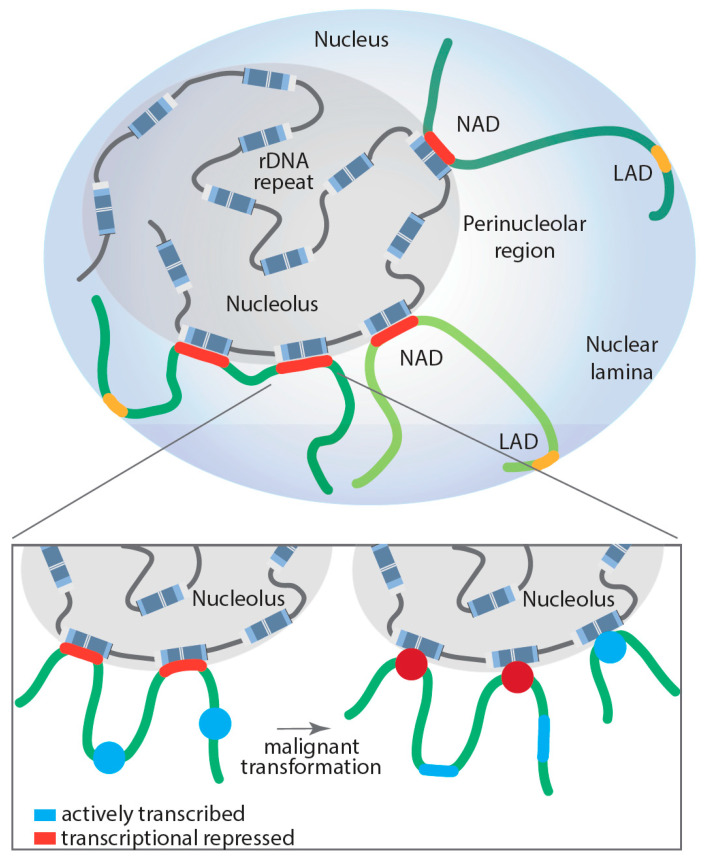
Proposed model of dynamically regulated long-range rDNA–NAD interactions during malignant transformation and their impact on Pol II-dependent transcription at associated loci. Lamina-associated domain (LAD); nucleolus associated domain (NAD); red: transcriptionally repressed Pol II-dependent genes; blue: actively transcribed Pol II-dependent genes.

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
