# Peer review of "The Ribosomal Gene Loci—The Power behind the Throne"

_genes, 2021, doi:10.3390/genes12050763_

Round 1

Reviewer 1 Report

This is a well written and engaging review on a topic that is timely and relatively underrepresented in the current literature. The only concern that I have is lack of acknowledgment that some non-RiBi consequences of perturbing nucleolar TFs or Pol1 activity may also be indirect (cell cycle arrest?, non-Pol1-related functions of those TFs). A discussion as to why such indirect effects are unlikely to explain the observed effects on differentiation on genomic stability could be useful.      

Reviewer 2 Report

Review of “The ribosomal gene loci - the power behind the throne”

This is a very complete discussion of the roles of the nucleolus in cellular biology. The authors have touched on nearly every base and are to be commended for the comprehensiveness of the work. However, there are some areas that need to be addressed.

  1. Loci is plural, the “powers” behind the throne?
  2. It is not clear if nucleoli form around ribosomal loci (repeats) that are actively transcribed…See papers by Brian McStay.
  3. Run on sentence lines 21-24; 24-27; and elsewhere
  4. Is it the ribosome repeats or the structure/organization of the nucleolus that is important?
  5. It is not clear if one can consider the “nucleolus” as a subnuclear domain of chromatin organization. Perhaps a subnuclear structure that affects the organization of non-ribosomal chromatin.
  6. Line 46. irregularly spaced no irregular spaced. However, I am not sure if Miller’s laboratory reported the presence of nucleosomes along the amplified ribosomal repeats of Xenopus
  7. Line 52. “79 ribosomal proteins transcribed by RNA polymerase II”. Ribosomal proteins are translated from mRNAs synthesized by Pol II.
  8. The statements on lines 72-74 require references.
  9. Line 76 and elsewhere. UBFT? I believe UBTF is correct. See https://www.genecards.org/cgi-bin/carddisp.pl?gene=UBTF
  10. Line 92. “ inter-subunit bridge elements that establish the binding interface linking the small and large ribosomal subunits” This should be explained in greater detail.
  11. Lines 113-121. To my knowledge, no one has ever demonstrated binding of SL1 to the CPE (-30-+1). Also, three laboratories demonstrated the binding of UBF upstream of the CPE, bridging between the CPE and UCE. Tjian’s laboratory reported that SL1 extended the UBF or UBTF, as it is now called, in the UCE. Rothblum’s lab demonstrated that SL1 bound to the UCE.
  12. Several laboratories have reported additional interactions between Pol I and the components of the committed template, g. PAF53 with UBTF.
  13. Line 122. “regulatory serine phosphorylation sites”
  14. Line 145. I believe the original identification of two forms of Pol I was due to the chromatographic separation of one form containing A34 and A49 from the second form that lacked those components. It is true that Rrn3 is required for transcription initiation, but it is not clear if that was the defining factor in this situation.
  15. Line 264. Do the authors mean cohesion or cohesin?
  16. Section 6. Was Engelke the first to report the association of genes transcribed by Pol III with the yeast nucleolus?
  17. Line 294. “…but did identified…”
  18. The description of NAD sequences is quite interesting. Did the two groups (references 9 and 11) use the same cell lines/tissues?
  19. Line 299. “Analysis of RNA coding genes…” Also, if this is the case, wouldn’t one expect to find Pol III transcription components and markers for active genes at the NAD’s as well as discussed starting on line 308.
  20. Paragraph, lines 388-399. It is not clear if the rate of rRNA synthesis can be correlated to malignancy. While it may be true that some tumors divide more rapidly than their surrounding, normal cells. They may not divide more rapidly than other normally dividing cells. And many tumor cells divide less rapidly than “normal” fibroblasts. An interesting model would be that they fail to respond to “No growth” signals or are over sensitive to “growth” signals.

Numerous grammatical errors that need to be corrected:

  For example:

  1. Run on sentences lines 21-24; 24-27; 282-297; 314-316. And others.
  2. Line 61, “In humans the rDNA genes are arranged in a head-to-tail orientation forming repetitive repeat arrays organized in the nucleolar organizer regions (NOR) at the short arm of the 5 acrocentric chromosomes.” “In humans,” Also do you have repetitive repeat arrays or repetitive arrays?
  3. Line 63, “precise organization and exact number of repeats is species”… organization and exact number are species…..
  4. Lines 271-273, 279-283, 361-363. These are awkward sentences and should be rewritten. There are numerous other examples.

Other comments:

Reference 1. Miller and Beatty did not report nucleosomes within the transcribed portion of the repeat.

Reference 18. Two other labs identified Pol I promoters in the mammalian spacer before Grummt’s lab:

  1. Rothblum’s lab demonstrated it in vitro
  2. Harrington’s lab demonstrated it in vivo.
  3. I believe that Zomerdijk’s most recent paper on the identification of the fourth TAF of SL1 is relevant.

Reference 30 has not been reproduced. Reference 29 is unnecessary.
